# Research on Coordination in a Dual-Channel Green Supply Chain under Live Streaming Mode

**Tianwen Chen [1,*], Ronghu Zhou [1], Changqing Liu [2,*] and Xiang Xu [3]**

1   Economic and Trade Department, Yancheng Polytechnic College, No. 285 Jiefang South Road,
    Yancheng 224005, China
2   Yancheng Institute of Technology, College of Marine and Biological Engineering, Yancheng 224051, China
3   School of Management Science and Engineering, Nanjing University of Finance and Economics,
    Nanjing 210023, China
*   Correspondence: 2015180414@yctei.edu.cn (T.C.); liuchangqing@ycit.edu.cn (C.L.);
    Tel.: +86-182-5225-7618 (T.C.)

**Abstract:** In this paper, we study the coordination issue in a dual-channel green supply chain with one manufacturer and one retailer. The demand in the traditional channel is assumed to be dependent on retail price, sales effort and green degree. Due to the characteristic of live broadcast selling, the demand in the direct channel is assumed to be dependent on price and discount. On the basis of analyzing price, sales effort and green degree strategies in the supply chain under the centralized model, two decentralized models and two coordination models are presented. Moreover, we prove the feasibility of sharing the R&D costs of the green degree and sales effort costs of the advertisement (CS-GS) contract through bargaining problems achieving a win-win situation, but the revenue sharing and wholesale price (RSC) contract commonly used cannot efficiently coordinate the supply chain. Finally, numerical analysis is given to show the impacts of coordination contracts on the supply chain's performance as well as the impacts of parameters on profits and decisions in the four models. It reveals that the CS-GS contract can not only help to improve the green degree and the price of the product, but also improve the profitability of all supply chain members.

**Keywords:** coordination; bargaining problem; green degree; sales effort

## 1. Introduction

With the development of the economy and the progress of science and technology, the problems of energy shortage, environmental pollution and the rapid development of human society have become more and more prominent and gradually become a bottleneck that restricts the further development of our society. In the past decades, environmental pollution and control measures have attracted wide attention all over the world. The issue is being closely concerned not only by governments but also by consumers. The U.S. Environmental Protection Agency, for example, encourages consumers to buy eco-friendly products and provides a web-based calculation tool to calculate the carbon emissions of each product [1]. With environmental friendliness and environmental protection being advocated globally, people have more and more demand for green products than before as their environmental awareness has increased in recent years.

Meanwhile, the rise of live-streaming selling has revolutionized the retail industry, which is rapidly becoming a major marketing tool in resale and agency sale platforms, especially after the outbreak of COVID-19. According to a report on China's live streaming industry, the number of live streaming users in China has reached 635 million by the end of 2021, accounting for 61.5% of total Internet users, with 464 million e-commerce live streaming users [2]. The year 2021 is one of the most significant years for live-streaming commerce, as many enterprises and departments have started to sell their products in live broadcast rooms. Especially in China, Taobao live streaming often promotes an event

by inviting internet celebrities to cooperate with brands to broadcast the products. This has been a successful way to increase conversion rates for many brands [3]. This paper is motivated by the reality of green products selling in live streaming rooms.

With the changeable and complex business environment, the supply chain management is becoming more and more difficult. Since the 1950s, many academic works have focused on the supply chain coordination issue [4–7]. A supply chain usually consists of different entities which make decisions individually; thus, information asymmetry and double marginalization will easily lead to lower overall system performance [8,9]. In order to manage the supply chain effectively, supply chain coordination between different members is necessary. Nowadays, there are many methods to test and verify the best policy which is helpful to win ideal profit. Derivative and game theory are widely used to solve this question. The game theory approach was regarded as a key study by many kinds of literature, which combine mathematical models and coordinate supply chains by contract mechanisms to achieve optimal efficiency [10–14]. The Supply chain coordination mechanism has two very important objectives: the first is to improve the overall efficiency and benefits of the supply chain when making decentralized decisions, and the second is that all members share the risks through the contracts [15].

The contributions of this paper are as follows: Firstly, we establish a two-echelon green supply chain that provides products through two channels and makes optimal decisions (as it is an entire system). Secondly, we build two decentralized models without coordinates, as expressed by the Stackelberg game, which is a manufacturer-dominant scenario and a retailer-dominant scenario. In these two decentralized models, the manufacturer and the retailer are two independent entities deriving their own profit, leading to double marginalization. To coordinate the supply chain, we propose and compare two coordination models, then prove the feasibility of the CS-GS contract that can help the supply chain achieve the same profits as that in the centralized model. Finally, we allocate the extra profit obtained by coordination through bargaining problems [16], which improves members' profits considering their bargaining power in the supply chain.

The structure of this paper is as follows. In Section 2, we review the related literature. In Section 3, model descriptions, notations and the integrated benchmark are introduced. In Section 4, we establish two decentralized models and analyze the optimal decisions made by each member. In Section 5, two coordination models are proposed and compared. On the basis of the CS-GS model, we talk about bargaining problems. In Section 6, numerical analysis is provided to verify the effectiveness of the coordination contracts and analyze the sensitivity of parameters. Finally, some conclusions and further insights are summarized in Section 7.

## 2. Related Literature

### 2.1. Research on Green Supply Chain

With the rise of a low-carbon economy and green GDP, the theory and practice of green supply chains have attracted more and more attention from scholars and market managers.

Some previous studies assumed demand as a function of the green degree to investigate the green supply chain. Zhang and Liu [17] considered a three-level green supply chain system, and the demand was related to the green degree of the product. A revenue sharing mechanism, Shapley value method coordination mechanism and asymmetric Nash negotiation mechanism were proposed to motivate supply chain members to cooperate in producing and marketing green products. Basiri and Heydari [18] investigated a two-stage green channel supply chain, which planned to release a new substitutable green product beside the current traditional product. Demand for both products was a function of the retail price, green quality and sales effort. Gao and Zhang [19] studied the pricing, green degree and sales effort decisions for a two-echelon green supply chain in which the demand is a linear form of them described by an uncertain variable. Xin et al. [20] established a two-echelon green product supply chain in which the demand for the green product is price, greenness and time sensitivity, so as to study the purchasing strategies, pricing

decisions and incentive mechanism for two selling periods. Jian et al. [21] built a green closed-loop supply chain involving a green product manufacturer and a retailer, and the demand was a function of the price, green degree and sales effort. The manufacturer was responsible for the green R&D costs and the recycling investment; at the same time, the retailer was responsible for the investment in the sales effort. Zong et al. [22] found that some manufacturers tend to misreport the green degree of their products to reduce the cost of energy conservation and emission reduction. They studied the impacts of misreporting behavior on the green supply chain performance and find that it would reduce the profits of the supply chain and the retailer.

The effect of consumer environment awareness on decisions is another research stream. Jiang and Yuan [23] provided decision-making suggestions from pricing points and green production, considering consumer environmental awareness. They built a prefabricated construction supply chain including a prefabricated company and a manufacturer in which the demand is a linear function related to total carbon reduction and proposed the cost-sharing contract and the two-part tariff to optimize the supply chain's profits.

Some scholars have also investigated the government's influence on the green supply chain. Li et al. [24] established a model of cooperative emission reduction involving the government, manufacturer, and retailer under different power structures. They found that the government can develop differentiated subsidy schemes to achieve Pareto optimality based on different game strategies and revenue-sharing agreements by enterprises. Gao et al. [25] focused on approaches to coordinate a dual-channel green supply chain composed of a manufacturer, a government and a retailer. The government provided a green standard for the manufacturer, and the manufacturer can obtain a subsidy from the government when the products satisfied the standard.

This paper is closely related to the work of Basiri and Heydari [18] and Jian et al. [21]. The demand in the offline channel is assumed to be dependent on the price, green degree and sales effort. Basiri and Heydari [18] considered the supply chain coordination issue in which the manufacturer prepared to put a new substitutable green product into the market and two types of products were sold on a single channel. Jian et al. [21] studied the impacts of the manufacturer's fairness concern behavior on the members and proposed profit-sharing contracts to coordinate the green closed-loop supply chain with fairness concerns. This paper considers a dual-channel supply chain with a single type of product, in which the manufacturer sells the products in two channels and pays for the R&D costs for the green degree. The CS-GS contract we proposed can fully coordinate the supply chain and improve the degree of the product.

### 2.2. Research on Supply Chain Coordination

Relevant research reveals that different entities in the supply chain driving their own benefits without considering the whole supply chain are harmful to the entire system, leading to double marginalization. Many coordination contracts have been studied in order to coordinate the supply chain and improve the interests of all members under decentralized decision-making models, such as wholesale price contacts [26], revenue sharing contacts [27], buy-back contracts [28,29], quantity discount contacts [30], sales rebate contracts [31] and so on.

The revenue-sharing contracts were widely investigated in previous studies to coordinate the supply chain. Peng et al. [32] analyzed a two-echelon supply chain system composed of multiple suppliers and multiple retailers, so as to propose a revenue-sharing (SRS) contract to coordinate the members. Moreover, they derived the conditions of the SRS contract achieving a win-win situation. Avinadav et al. [33] considered a manufacturer who launched his app to consumers via an online retailer with a revenue-sharing contract, in which the receive signals were related to the uncertain demand and the demand was a random variable affected by both selling price and quality improvement.

The buy-back contracts have been extended in order to coordinate the supply chain in more and more complex situations [28]. He and Zhao [34] considered an integrated

model and designed a returns policy between the manufacturer and the retailer to create a situation for both sides to win. Wang et al. [35] proposed a buy-back contract with return cost to coordinate a dual channel system comprising a brand-owned direct channel and a retail outsourcing channel. Then studied supply chain sustainability and examine the effect of two key influencing factors (price competition and demand uncertainty). Mojtaba et al. [36] developed a buy-back coordination mechanism to coordinate a supply chain consisting of one retailer and one manufacturer, which is related to expire items instead of unsold products different from conventional buy-back contracts, and proved that the proposed strategy is economically feasible when the revenue of each reused product excepted the saving on its disposing cost is greater than its reprocessing cost. Lee and Rhee [37] developed strategies by using five commonly used contracts (revenue-sharing, buybacks, quantity flexibility, quantity discount, and two-part tariff) to incentivize retailers to make decisions for the benefit of the entire system, and proposed strategies to coordinate a supply chain in which the retailer allowed buyers to return goods after inspection and to resell them after partial consumption.

Another stream of research focused on some new contracts. Yang et al. [38] compared three coordination policies to determine the maximum profit under the stock-dependent demand rate situation, and they are the credit period policy, the quantity discount policy and the centralized supply chain policy. Hosseini-Motlagh et al. [39] proposed a profit surplus distribution (PSD) mechanism to coordinate a two-stage supply chain, which includes a retailer investing in sales efforts and a population of manufacturers investing in innovation efforts. Tsou [40] tried to find out the criteria for inventory management strategy decisions to reduce the inventory risk and increase profitability and used three tests to explore the best decisions which are the sequential probability ratio test, cumulative sum chart test and auto-regression test. Wang et al. [41] established a two-echelon supply chain including a supplier and a manufacturer who is overconfident about the yield. The optimal production and ordering decisions were examined in four contract models: wholesale price contract with complete rationality, wholesale price contract with overconfidence, option contract with complete rationality, and option contract with overconfidence.

The final purpose of the research mentioned above was to determine the maximum profitability of the supply chain. However, few researchers considered the bargaining power of members in the supply chain. By referring to the research about five commonly used contracts and extended contracts, two new joint contracts are proposed in this paper to coordinate the supply chain. It is verified that the supply chain can obtain the maximum profits from the CS-GS coordination contract. Moreover, we discussed the extra profit allocation by bargaining problem achieving a win-win situation.

### 2.3. Research on Supply Chain Models and Decisions

The third area of research centered on decision-making and model-building. In terms of decision-making, some decisions were usually studied in the related literature, such as price, production, ordering, inventory, and so on. Most of the literature analyzed the decision-making problems by a game model with different power structures. SeyedEsfahani et al. [42] considered vertical co-op advertising along with pricing decisions in a supply chain and established three non-cooperative games including Nash, Stackelberg-manufacturer and Stackelberg-retailer and one cooperative game. Zhai and Cheng [43] constructed a model to study how PTH (Production Time Hedging) affects the retailer's QDLT (Quoted Delivery Lead-Time) decision and the supply chain performance. They established four models, the centralized model, the Nash model, the manufacturer-led Stackelberg model and the retailer-led Stackelberg model to derive closed-form results on the optimal hedging and QDLT decisions. Zhong et al. [44] analyzed a two-stage Stackelberg game model, in which both the demand and yield are uncertain. They studied and compared the production and ordering decisions under two different models, and proposed a subsidy mechanism to achieve Pareto-improvement.

Some scholars considered the supply chain with more members and parties. Zhao and Wang [45] established three different game structures, namely Manufacturer-leader Stackelberg, Retailer-leader Stackelberg and Vertical Nash. Expected value models are developed to determine the optimal pricing and retail service strategies in a supply chain with one manufacturer and two retailers. Hanh et al. [46] studied price and production decisions in a three-stage supply chain including multiple suppliers, multiple manufacturers and multiple retailers where the suppliers are the leaders and the retailers are the followers, and the demand is price sensitive. Through the Salop spatial model and Nash game approach to minimize the double marginalization effect. Zhao and Ma [47] built a three-party game supply chain model involving a battery manufacturer, a car manufacturer, and a third-party recycler. The retail price decisions of new energy vehicles, new batteries and third-party recyclers are analyzed under different decision models. Under the decentralized model, the profits of battery manufacturers and third-party recyclers are affected by the retail price of used batteries, while the car manufacturers are not.

Another group of literature relevant to our research established the supply chain by expanding the channel for sales. Wang et al. [48] established a dual-channel tourism supply chain model with OTA (online travel agencies) channel participation and provided a pricing decision model regarding the effect of service quality by TPP (tourism product providers) leading the Stackelberg game. Wang et al. [49] investigated the optimal retailer price, delivery distance and allowable return period in a dual-channel supply chain by the Stackelberg game model, in which the manufacturer sells the products through online and offline channels to consumers and the demand is a linear form of the prices in two selling channels.

The previous works on the supply chain models and decisions mentioned above usually concerned the assumption that the demand is price sensitive and that the products are sold via traditional channels. We consider a new type of selling channel called live streaming rooms beside the traditional selling channel, which is becoming very popular in sales platforms, especially after the outbreak of COVID-19. In addition, we assume the demand for live-streaming rooms is related not only to the price but also to the discount because of the characteristics of the live-streaming rooms.

## 3. Model Descriptions, Notations and the Integrated Benchmark

### 3.1. Model Descriptions, Notations and Assumptions

We consider a dual-channel green supply chain that includes one retailer and one manufacturer. As online live-streaming rooms are becoming all the rage on the Internet in China, some people find this type of selling place appealing, the manufacturer sells the products in live-streaming rooms to such consumers directly and the retailer sells the products to other consumers via the traditional channel. We assume that the production is the demand. The demand in the direct channel is assumed to be dependent on price and discount; the demand in the traditional channel is assumed to be dependent on retail price, sales effort and green degree. The manufacturer needs to decide the green degree of the product. The retailer needs to decide the price and the sales effort of the advertisement. Nomenclature part are used to formulate the supply chain model discussed in this paper.

**Assumption 1.** *$P > w > c > 0$, which makes sure that the manufacturer and the retailer are willing to participate in this supply chain, $\widetilde{p} > c > 0$ which makes sure that the manufacturer can obtain profits from the live streaming rooms.*

**Assumption 2.** *We assume the customers are of two types, one that is accustomed to consuming offline via traditional channels, and another that is attracted by the discount and convenience in the live streaming rooms. The demand in the traditional channel is assumed to be $D_1(p, e, \theta) = z\alpha - \beta p + ke + \gamma\theta$ [50], the demand in live streaming rooms is assumed to be $D_2(p) = (1 - z)\alpha - \beta\widetilde{p}$. Here $\widetilde{p} = (1 - d)p$, $D_1 > 0$, $D_2 > 0$, z is the proportion of the market in the traditional channel, which is a nonnegative variable.*

**Assumption 3.** *α, β, k, d and γ are independent and nonnegative.*

**Assumption 4.** *The function of R&D costs is to improve the green degree of the product.* $g(\theta) = \frac{1}{2}\tau\theta^2$ [51].

**Assumption 5.** *The function of sales effort costs of the advertisement.* $t(e) = \frac{1}{2}be^2$ [52]. *The two functions are convex, it is obvious that* $g(0) = 0, g'(\theta) = \tau\theta > 0, g''(\theta) = \tau > 0, t(0) = 0, t'(e) = be > 0, t''(e) = b > 0.$

Profit function

The supplier's, retailer's and the whole supply chain's profit can be expressed as follows:

$$\pi_m(\theta) = (w - c)D_1(p, e, \theta) + (\widetilde{p} - c)D_2(p) - g(\theta) \tag{1}$$

The first part is the manufacturer's profit from the wholesales, the second part is the manufacturer's profit from live streaming rooms and the third part is the manufacturer's green costs.

$$\pi_r(e, p) = (p - w)D_1(p, e, \theta) - t(e) \tag{2}$$

The first part is the retailer's selling revenue, and the second part is the retailer's advertisement cost.

$$\pi_c(\theta, e, p) = (p - c)D_1(p, e, \theta) + (\widetilde{p} - c)D_2(p) - t(e) - g(\theta) \tag{3}$$

The first part is the revenue from sales in traditional channel, the second part is the revenue from sales in live streaming rooms, the third part is the advertisement costs and the last is the green R&D costs.

*3.2. The Integrated Benchmark*

To establish a performance benchmark, we first analyze the problem of an integrated supply chain. The integrated firm tries to maximize its profit, denoted as

$$\pi_c(\theta, e, p) = (p - c)(z\alpha - \beta p + ke + \gamma\theta) + [(1 - d)p - c][(1 - z)\alpha - \beta(1 - d)p] - \frac{1}{2}\tau\theta^2 - \frac{1}{2}be^2 \tag{4}$$

**Proposition 1.** *The expected profit* $\pi_c(\theta, e, p)$ *is jointly concave in* $(\theta, e, p)$. *If* $-2b\tau\beta d^2 + 4b\tau\beta d + \tau k^2 + b\gamma^2 - 4b\tau\beta < 0$, *the optimal retail price, the optimal sales effort of the advertisement, and the green degrees of the product, donated as* $(\theta^*, e^*, p^*)$, *respectively, are given as follows:*

$$\theta^* = \frac{k\tau(2\beta c - \alpha + \alpha d - 3\beta cd - \alpha dz + 2\beta cd^2)}{-2b\tau\beta d^2 + 4b\tau\beta d + \tau k^2 + b\gamma^2 - 4b\tau\beta} \tag{5}$$

$$e^* = \frac{b\gamma(2\beta c - \alpha + \alpha d - 3\beta cd - \alpha dz + 2\beta cd^2)}{-2b\tau\beta d^2 + 4b\tau\beta d + \tau k^2 + b\gamma^2 - 4b\tau\beta} \tag{6}$$

$$p^* = \frac{c\tau k^2 + bc\gamma^2 - \alpha b\tau - 2bc\tau\beta + \alpha bd\tau - \alpha bd\tau z + bcd\tau\beta}{-2b\tau\beta d^2 + 4b\tau\beta d + \tau k^2 + b\gamma^2 - 4b\tau\beta} \tag{7}$$

**Proof of Proposition 1.** The first-order of Equation (4) with respect to $\theta, e, p$, respectively, are

$$\frac{\partial \pi_c(\theta, e, p)}{\partial \theta} = \gamma(p - c) - \tau\theta$$

$$\frac{\partial \pi_c(\theta, e, p)}{\partial e} = k(p - c) - be$$

$$\frac{\partial \pi_c(\theta, e, p)}{\partial p} = z\alpha - 2\beta p + ke + \gamma\theta + \beta c + (1 - d)[(1 - z)\alpha - 2\beta(1 - d)p + \beta c]$$

Then the Hessian matrix is

$$
H_1 = \begin{bmatrix}
\dfrac{\partial^2 \pi_c(\theta, e, p)}{\partial \theta^2} & \dfrac{\partial^2 \pi_c(\theta, e, p)}{\partial \theta \partial e} & \dfrac{\partial^2 \pi_c(\theta, e, p)}{\partial \theta \partial p} \\[2mm]
\dfrac{\partial^2 \pi_c(\theta, e, p)}{\partial e \partial \theta} & \dfrac{\partial^2 \pi_c(\theta, e, p)}{\partial e^2} & \dfrac{\partial^2 \pi_c(\theta, e, p)}{\partial e \partial p} \\[2mm]
\dfrac{\partial^2 \pi_c(\theta, e, p)}{\partial p \partial \theta} & \dfrac{\partial^2 \pi_c(\theta, e, p)}{\partial p \partial e} & \dfrac{\partial^2 \pi_c(\theta, e, p)}{\partial p^2}
\end{bmatrix} = \begin{bmatrix}
-\tau & 0 & \gamma \\
0 & -b & k \\
\gamma & k & -2\beta - 2\beta(1-d)^2
\end{bmatrix}
$$

According to the assumption proposed before, when $-2b\tau\beta d^2 + 4b\tau\beta d + \tau k^2 + b\gamma^2 - 4b\tau\beta < 0$ and $2b\beta + 2b\beta(1-d)^2 - k^2 > 0$, it is easy to check that the Hessian matrix is a negative define matrix, which implies that $\pi_c(\theta, e, p)$ is jointly concave in $(\theta^*, e^*, p^*)$, and the maximum $(\theta^*, e^*, p^*)$ is uniquely solved by

$$
\gamma(p - c) - \tau\theta = 0
$$

$$
k(p - c) - be = 0
$$

$$
\alpha - 2\beta p + ke + \gamma\theta + \beta c + (1-d)[(1-z)\alpha - 2\beta(1-d)p + \beta c] = 0
$$

The unique optimal $\theta^*$, $e^*$, $p^*$ should satisfy the first-order conditions. Hence, we have the proposition.

The maximal expected profit of the integrated firm is

$$
\begin{aligned}
\pi_c^* = \pi_c(\theta^*, e^*, p^*) &= Max\,\pi_c(\theta, e, p) \\
&= \frac{\begin{aligned}
&-b\tau\alpha^2 d^2 z^2 + 2b\tau\alpha^2 d^2 z - b\tau\alpha^2 d^2 - 2b\tau\alpha^2 dz + 2b\tau\alpha^2 d - b\tau\alpha^2 + 2b\tau\alpha\beta cd^2 z + 2b\tau\alpha\beta cd^2 - 4b\tau\alpha\beta cdz - 2b\tau\alpha\beta cd \\
&+ 4b\tau\alpha\beta c + 2b\alpha cd\gamma^2 z - 2b\alpha cd\gamma^2 + 2\tau\alpha cdk^2 z - 2\tau\alpha cdk^2 - b\tau\beta^2 c^2 d^2 + 4b\tau\beta^2 c^2 d - 4b\tau\beta^2 c^2 - 2b\beta c^2 d^2\gamma^2 \\
&-2\tau\beta c^2 d^2 k^2 + 2b\beta c^2 d\gamma^2 + 2\tau\beta c^2 dk^2
\end{aligned}}{2(-2b\tau\beta d^2 + 4b\tau\beta d + \tau k^2 + b\gamma^2 - 4b\tau\beta)}
\end{aligned} \tag{8}
$$

$\square$

## 4. The Decentralized Model

In this section, we consider the decentralized supply chain that involves two self-profit-maximizing firms, which is assumed that there is no coordination in it. We formulated the decentralized problem as a Stackelberg game, and derive the equilibrium solutions. Then we compare the performances of the supply chain in these models.

### 4.1. Manufacturer-Dominant Decentralized Model (MD Model)

Firstly, we suppose the manufacturer is a leader, and the retailer is the follower. In the MD model, given the earlier decision $(e, p)$ made by the retailer, the manufacturer's optimal green degree of the product can be achieved. We first derive the retailer's reaction function as follows:

$$
\pi_r(e, p) = (p - w)(z\alpha - \beta p + ke + \gamma\theta) - \frac{1}{2} be^2 \tag{9}
$$

$$
\frac{\partial \pi_r(e, p)}{\partial e} = k(p - w) - be \qquad \frac{\partial \pi_r(e, p)}{\partial p} = (z\alpha - \beta p + ke + \gamma\theta) - \beta(p - w)
$$

$$
\begin{cases}
e_d^{md*} = \dfrac{k(\alpha z - \beta w + \theta\gamma)}{2\beta b - k^2} \\[3mm]
p_d^{md*} = \dfrac{zb\alpha + b\theta\gamma + b\beta w - k^2 w}{2\beta b - k^2}
\end{cases} \tag{10}
$$

Then, after achieving the retailer's reaction function, the manufacturer sets the optimal green degree of the product.

**Proposition 2.** *In the MD model, the manufacturer's optimal wholesale price, the retailer's optimal price and the optimal sales effort are denoted as* $\left(\theta_d^{md*}, e_d^{md*}, p_d^{md*}\right)$.

$$e_d^{md*} = \frac{k\left(\alpha z - \beta w + \theta_d^{md*}\gamma\right)}{2\beta b - k^2} \tag{11}$$

$$p_d^{md*} = \frac{zb\alpha + b\theta_d^{md*}\gamma + b\beta w - k^2 w}{2\beta b - k^2} \tag{12}$$

$$\theta_d^{md*} = \frac{-\dfrac{b\,\beta\,\gamma\,(c-w)}{\sigma_1} + \dfrac{b\,\gamma\,(d-1)\left(\alpha\,(z-1) - \dfrac{\beta\,(d-1)\,\sigma_2}{\sigma_1}\right)}{\sigma_1} - \dfrac{b\,\beta\,\gamma\left(c + \dfrac{(d-1)\,\sigma_2}{\sigma_1}\right)(d-1)}{\sigma_1}}{\tau + \dfrac{2\,b^2\,\beta\,\gamma^2\,(d-1)^2}{\sigma_1^2}} \tag{13}$$

*where:*

$$\sigma_1 = 2\,b\,\beta - k^2 \qquad \sigma_2 = -w\,k^2 + b\,\beta\,w + \alpha\,b\,z$$

**Proof of Proposition 2.** Substituting Equation (10) into Equation (1) we have

$$\pi_m^{md}(\theta) = (w-c)\left(z\alpha - \beta p_d^{md*} + ke_d^{md*} + \gamma\theta\right) + \left[(1-d)p_d^{md*} - c\right]\left[(1-z)\alpha - \beta(1-d)p_d^{md*}\right] - \frac{1}{2}\tau\theta^2 \tag{14}$$

Taking the first derivative of Equation (14) with respect to θ, we obtain

$$\frac{\partial \pi_m^{md}(\theta)}{\partial \theta} = \frac{b\,\gamma\left(\alpha\,(z-1) - \dfrac{\beta\,(d-1)\,\sigma_2}{\sigma_1}\right)(d-1)}{\sigma_1} - (c-w)\left(\gamma + \frac{\gamma k^2}{\sigma_1} - \frac{b\beta\gamma}{\sigma_1}\right) - \tau\,\theta - \frac{b\,\beta\,\gamma\,(d-1)\left(c + \dfrac{(d-1)\,\sigma_2}{\sigma_1}\right)}{\sigma_1} = 0$$

where

$$\sigma_1 = 2\,b\,\beta - k^2 \qquad \sigma_2 = -w\,k^2 + b\,\beta\,w + \alpha\,b\,z + b\,\gamma\,\theta$$

Taking the second derivative of Equation (14) with respect to θ, we obtain

$$\frac{\partial^2 \pi_m^{md}(\theta)}{\partial \theta^2} = -\tau - \frac{2\,b^2\,\beta\,\gamma^2\,(d-1)^2}{\left(2\,b\,\beta - k^2\right)^2} < 0$$

Then we have unique $\theta_d^{md*}$ which achieves the manufacturer's optimal profit, and by integrating Equation (13) into Equation (10), we can determine the retailer's optimal decisions. Lastly, we can determine the maximal profit for the manufacturer and the retailer.

$$\pi_m^{md*} = Max\pi_m(\theta) = \pi_m\left(\theta_d^{md*}\right)$$

$$= \frac{\begin{array}{l} -2\tau\alpha^2b^2\beta d^2z^2 + 8\tau\alpha^2b^2\beta dz^2 - 4\tau\alpha^2b^2\beta dz - 6\tau\alpha^2b^2\beta z^2 + 4\tau\alpha^2b^2\beta z + \alpha^2b^2d^2\gamma^2z^2 - 2\alpha^2b^2d^2\gamma^2z + \alpha^2b^2d^2\gamma^2 - 2\alpha^2b^2d\gamma^2 \\ +4\alpha^2b^2d\gamma^2z - 2\alpha^2b^2d\gamma^2 + \alpha^2b^2\gamma^2z^2 - 2\alpha^2b^2\gamma^2z + \alpha^2b^2\gamma^2 - 2\tau\alpha^2bdk^2z^2 + 2\tau\alpha^2bdk^2z + 2\tau\alpha^2bk^2z^2 - 2\tau\alpha^2bk^2z \\ -4\tau\alpha b^2\beta^2cdz + 8\tau\alpha b^2\beta^2cz - 8\tau\alpha b^2\beta^2c - 4\tau\alpha b^2\beta^2d^2wz + 12\tau\alpha b^2\beta^2dwz - 4\tau\alpha b^2\beta^2dw - 4\tau\alpha b^2\beta^2wz + 4\tau\alpha b^2\beta^2w \\ +2\alpha b^2\beta cd^2\gamma^2z - 2\alpha b^2\beta cd^2\gamma^2 - 6\alpha b^2\beta cd\gamma^2z + 6\alpha b^2\beta cd\gamma^2 + 4\alpha b^2\beta c\gamma^2z - 4\alpha b^2\beta c\gamma^2 + 2\alpha b^2\beta d\gamma^2wz - 2\alpha b^2\beta d\gamma^2w \\ -2\alpha b^2\beta\gamma^2wz + 2\alpha b^2\beta\gamma^2w + 2\tau\alpha b\beta cdk^2z - 8\tau\alpha b\beta ck^2z + 8\tau\alpha b\beta ck^2 + 4\tau\alpha b\beta d^2k^2wz - 14\tau\alpha b\beta dk^2wz + 6\tau\alpha b\beta dk^2w \\ +8\tau\alpha b\beta k^2wz - 6\tau\alpha b\beta k^2w + 2\tau\alpha ck^4z - 2\tau\alpha ck^4 + 2\tau\alpha dk^4wz - 2\tau\alpha dk^4w - 2\tau\alpha k^4wz + 2\tau\alpha k^4w - 4\tau b^2\beta^3cdw + 8\tau b^2\beta^3c \\ -2\tau b^2\beta^3d^2w^2 + 4\tau b^2\beta^3dw^2 - 6\tau b^2\beta^3w^2 + b^2\beta^2c^2d^2\gamma^2 + 4b^2\beta^2cd^2\gamma^2w - 10b^2\beta^2cd\gamma^2w + 4b^2\beta^2c\gamma^2w - 4b^2\beta^2d^2\gamma^2w^2 \\ +8b^2\beta^2d\gamma^2w^2 - 3b^2\beta^2\gamma^2w^2 + 6\tau b\beta^2cdk^2w - 8\tau b\beta^2ck^2w + 4\tau b\beta^2d^2k^2w^2 - 8\tau b\beta^2dk^2w^2 + 6\tau b\beta^2k^2w^2 - 2\tau\beta cdk^4w \\ +2\tau\beta ck^4w - 2\tau\beta d^2k^4w^2 + 4\tau\beta dk^4w^2 - 2\tau\beta k^4w^2 \end{array}}{2\left(4\tau b^2\beta^2 + 2b^2\beta d^2\gamma^2 - 4b^2\beta d\gamma^2 + 2b^2\beta\gamma^2 - 4\tau b\beta k^2 + \tau k^4\right)} \tag{15}$$

$$\pi_r^{md*} = Max\pi_r(e,p) = \pi_r\left(e_d^{md*}, p_d^{md*}\right)$$

$$= \frac{b\left(2b\beta - k^2\right)\left(\begin{array}{l}\alpha bd\gamma^2 - \alpha b\gamma^2 + b\beta\gamma^2w + \alpha b\gamma^2z + 2b\beta^2\tau w - \beta k^2\tau w + \alpha k^2\tau z + 2b\beta d^2\gamma^2w - 2\alpha b\beta\tau z + b\beta cd\gamma^2 - 4b\beta d\gamma^2w \\ -\alpha bd\gamma^2z\end{array}\right)^2}{2\left(4\tau b^2\beta^2 + 2b^2\beta d^2\gamma^2 - 4b^2\beta d\gamma^2 + 2b^2\beta\gamma^2 - 4\tau b\beta k^2 + \tau k^4\right)^2} \tag{16}$$

It is obvious that $p_d^{md*} \neq p^*$, which indicates that wholesale-price-only contracts cannot coordinate the supply chain. Due to the double marginalization, $\pi_r^{md*} + \pi_m^{md*} = \pi_d^{md*} < \pi_c^*$, which means we cannot achieve the maximum profits in this decentralized supply chain under the manufacturer-dominant Stackelberg game. □

### 4.2. Retailer-Dominant Decentralized Model (RD Model)

Now we consider the manufacturer as a leader and the supplier as a follower. In the RD model, given the earlier decision $\theta$ made by the manufacturer, the retailer's optimal price and the optimal sales effort of the advertisement can be achieved. We first derive the retailer's reaction function as follows.

$$\pi_m(\theta) = (w-c)(z\alpha - \beta p + ke + \gamma\theta) + [(1-d)p - c][(1-z)\alpha - \beta(1-d)p] - \frac{1}{2}\tau\theta^2 \quad (17)$$

$$\frac{\partial \pi_m(\theta)}{\partial \theta} = -\tau\theta - \gamma(c-w) = 0$$

$$\frac{\partial^2 \pi_m(\theta)}{\partial \theta^2} = -\tau < 0$$

$$\theta_d^{rd*} = \frac{\gamma(w-c)}{\tau} \quad (18)$$

Then, after achieving the manufacturer's reaction function, the retailer sets the optimal price and the sales effort of the advertisement.

**Proposition 3.** *In the RD model, the manufacturer's optimal green degree of the product, the retailer's optimal price and the optimal sales effort are denoted as $\left(\theta_d^{rd*}, e_d^{rd*}, p_d^{rd*}\right)$ and*

$$\theta_d^{rd*} = \frac{\gamma(w-c)}{\tau} \quad (19)$$

$$e_d^{rd*} = \frac{k\left(-c\gamma^2 + \gamma^2 w - \beta\tau w + \alpha\tau z\right)}{\tau(2b\beta - k^2)} \quad (20)$$

$$p_d^{rd*} = \frac{b\gamma^2 w - bc\gamma^2 - k^2\tau w + b\beta\tau w + \alpha b\tau z}{\tau(2b\beta - k^2)} \quad (21)$$

**Proof of Proposition 3.** Substituting Equation (19) into Equation (2), taking the first derivative with respect to $e$ and $p$, we have

$$\pi_r^{rd}(e,p) = (p-w)(z\alpha - \beta p + ke + \frac{\gamma^2(w-c)}{\tau}) - \frac{1}{2}be^2 \quad (22)$$

Taking the first derivative of Equation (22) with respect to $e$ and $p$, we obtain

$$\frac{\partial \pi_r^{rd}(e,p)}{\partial e} = k(p-w) - be = 0$$

$$\frac{\partial \pi_r^{rd}(e,p)}{\partial p} = ek - \beta p + \alpha z - \beta(p-w) + \frac{\gamma^2(w-c)}{\tau} = 0$$

Then the Hessian matrix is

$$H_2 = \begin{bmatrix} \dfrac{\partial^2 \pi_r^{rd}(e,p)}{\partial e^2} & \dfrac{\partial^2 \pi_r^{rd}(e,p)}{\partial e\partial p} \\ \dfrac{\partial^2 \pi_r^{rd}(e,p)}{\partial p\partial e} & \dfrac{\partial^2 \pi_r^{rd}(e,p)}{\partial p^2} \end{bmatrix} = \begin{bmatrix} -b & k \\ k & -2\beta \end{bmatrix}$$

It is obvious that when $2\beta b - k^2 > 0$, the Hessian matrix is a negative define matrix, which implies that $\pi_r(e,p)$ is jointly concave in $\left(p_d^{rd*}, e_d^{rd*}\right)$, and the maximum $\left(p_d^{rd*}, e_d^{rd*}\right)$ is uniquely solved by

$$k(p-w) - be = 0$$

$$ek - \beta p + \alpha z - \beta(p-w) - \frac{\gamma^2(c-w)}{\tau} = 0$$

Lastly, we can calculate the maximal profit of the manufacturer and the retailer as follows.

$$\pi_r^{rd*} = Max\ \pi_r(e,p) = \pi_r\left(e_d^{rd*},\ p_d^{rd*}\right) = \frac{b\left(c\,\gamma^2 - \gamma^2\,w + \beta\,\tau\,w - \alpha\,\tau\,z\right)^2}{2\,\tau^2\left(2\,b\,\beta - k^2\right)} \tag{23}$$

$$\begin{aligned}
\pi_m^{rd*} = Max\ \pi_m(\theta) &= \pi_m\left(\theta_d^{rd*}\right) \\
&= \left(c + \frac{(d-1)\,\sigma_1}{\sigma_2}\right)\left(\alpha\,(z-1) - \frac{\beta\,(d-1)\,\sigma_1}{\sigma_2}\right) - \frac{\gamma^2\,(c-w)^2}{2\,\tau} \\
&\quad + \frac{b\,\beta\,(c-w)\left(c\,\gamma^2 - \gamma^2\,w + \beta\,\tau\,w - \alpha\,\tau\,z\right)}{\sigma_2}
\end{aligned} \tag{24}$$

where:

$$\sigma_1 = b\,\gamma^2\,w - b\,c\,\gamma^2 - k^2\,\tau\,w + b\,\beta\,\tau\,w + \alpha\,b\,\tau\,z$$

$$\sigma_2 = \tau\left(2\,b\,\beta - k^2\right)$$

It is obvious that $p_d^{rd*} \neq p^*$ indicates that the supply chain in the decentralized model will not make decisions in the same way as it makes optimal decisions when it is an integrated system. Due to the double marginalization, $\pi_m^{rd*} + \pi_r^{rd*} = \pi_d^{rd*} < \pi_c^*$, which means we cannot achieve the maximum profits in this decentralized supply chain under the retailer-dominant Stackelberg game. □

## 5. Supply Chain Coordination

When the manufacturer and the retailer are two independent entities, they will try to maximize their own expected profits without thinking about other members of this supply chain. Hence, if we want to let the retailer and the manufacturer make decisions in the same way as an integrated benchmark, which means maximizing the total profit of the supply chain, we should offer proper contracts to encourage and constrain them and then the supply chain's coordination will be achieved.

### 5.1. RSC Contract

From above, we know that the optimal supply-chain profit can be achieved by choosing the green degree, the sales effort of the advertisement and the price of the product as $(\theta^*, e^*, p^*)$, and an arbitrary allocation of the optimal profit can be achieved by varying $w$. The next question is how to decide the wholesale price that the manufacturer asked for. In a wholesale price-only contract, the manufacturer charges the retailer a wholesale price higher than its marginal cost and influences the retailer's price decision. In the RSC model, we combine wholesale price contracts with the revenue sharing contract to coordinate the supply chain. We denote such a contract as $(w,\phi)$, $0 < \phi < 1$.

In the RSC model, to encourage the retailer to order more products at the beginning, the manufacturer wholesale the products at a lower price $w$, and the retailer shares $\phi$ proportion of revenue with the manufacturer at the end of the selling period to make up for the loss of lower wholesale price. The profits of the manufacturer and the retailer are as follows:

$$\pi_m^{RSC}(\theta) = (w-c)(z\alpha - \beta p + ke + \gamma\theta) + \phi(p-w)(z\alpha - \beta p + ke + \gamma\theta) + [(1-d)p - c][(1-z)\alpha - \beta(1-d)p] - \frac{1}{2}\tau\theta^2 \tag{25}$$

$$\pi_r^{RSC}(e,p) = (1-\phi)(p-w)(z\alpha - \beta p + ke + \gamma\theta) - \frac{1}{2}be^2 \tag{26}$$

**Proposition 4.** *The RSC contract cannot coordinate the supply chain effectively. The optimal price, optimal green degree and optimal sales effort derived in the RSC model cannot achieve the corresponding decisions in the centralized model simultaneously by $w$ and $\phi$.*

**Proof of Proposition 4.** In a similar way, we formulate the RSC model as a Stackelberg game and suppose the retailer is the leader. Firstly, we derive the retailer's reaction function as follows.

$$\frac{\partial \pi_m^{RSC}(\theta)}{\partial \theta} = \gamma\,\phi\,(p-w) - \gamma\,(c-w) - \tau\,\theta = 0$$

$$\frac{\partial^2 \pi_m^{RSC}(\theta)}{\partial \theta^2} = -\tau < 0 \tag{27}$$

$$\theta_d^{RSC*} = \frac{\gamma\,\phi\,(p-w) - \gamma\,(w-c)}{\tau}$$

Substituting Equation (27) into Equation (25), taking the first derivative with respect to $e$ and $p$, we have

$$\pi_r^{RSC}(e, p) = (p - w)\left(z\alpha - \beta p + ke + \frac{\gamma^2 \phi (p - w) - \gamma (w - c)}{\tau}\right) - \frac{1}{2}be^2 \quad (28)$$

Taking the first derivative of Equation (28) with respect to $e$ and $p$, we obtain

$$\frac{\partial \pi_r^{RSC}(e, p)}{\partial e} = -be - k(p - w)(\phi - 1)$$

$$\frac{\partial \pi_r^{RSC}(e, p)}{\partial p}\left(\beta - \frac{\gamma^2 \phi}{\tau}\right)(p - w)(\phi - 1) - (\phi - 1)\left(ek - \beta p + \alpha z - \frac{\gamma(\gamma(c - w) - \gamma\phi(p - w))}{\tau}\right)$$

Then the Hessian matrix is

$$H_2 = \begin{bmatrix} \dfrac{\partial^2 \pi_r^{rd}(e, p)}{\partial e^2} & \dfrac{\partial^2 \pi_r^{rd}(e, p)}{\partial e \partial p} \\ \dfrac{\partial^2 \pi_r^{rd}(e, p)}{\partial p \partial e} & \dfrac{\partial^2 \pi_r^{rd}(e, p)}{\partial p^2} \end{bmatrix} = \begin{bmatrix} -b & k(1 - \phi) \\ k(1 - \phi) & -2\left(\beta - \frac{\gamma^2 \phi}{\tau}\right)(1 - \phi) \end{bmatrix}$$

It is obvious that when $2b\left(\beta - \frac{\gamma^2 \phi}{\tau}\right) - k^2(1 - \phi) > 0$, the Hessian matrix is a negatively defined matrix, which implies that $\pi_r^{RSC}(e, p)$ is jointly concave in $(e_d^{RSC*}, p_d^{RSC*})$, and the maximum $(e_d^{RSC*}, p_d^{RSC*})$ is solved uniquely by

$$-be - k(p - w)(\phi - 1) = 0$$

$$\left(\beta - \frac{\gamma^2 \phi}{\tau}\right)(p - w)(\phi - 1) - (\phi - 1)\left(ek - \beta p + \alpha z - \frac{\gamma(\gamma(c - w) - \gamma\phi(p - w))}{\tau}\right) = 0$$

The manufacturer's optimal green degree of the product, the retailer's optimal price and optimal sales effort, denoted as $\left(\theta_d^{RSC*}, e_d^{RSC*}, p_d^{RSC*}\right)$, are

$$\theta_d^{RSC*} = \frac{-\gamma(ck^2\tau - k^2\tau w + 2b\beta\tau w + bc\gamma^2\phi - b\gamma^2\phi w - ck^2\phi\tau + k^2\phi\tau w - 2b\beta c\tau - b\beta\phi\tau w + \alpha b\phi\tau z)}{\tau(k^2\tau - k^2\phi\tau - 2b\beta\tau + 2b\gamma^2\phi)} \quad (29)$$

$$e_d^{RSC*} = \frac{k(1 - \phi)(c\gamma^2 - \gamma^2 w + \beta\tau w - \alpha\tau z)}{k^2\tau - k^2\phi\tau - 2b\beta\tau + 2b\gamma^2\phi} \quad (30)$$

$$p_d^{RSC*} = \frac{-b\gamma^2 w + bc\gamma^2 + k^2\tau w - b\beta\tau w - \alpha b\tau z + 2b\gamma^2\phi w - k^2\phi\tau w}{k^2\tau - k^2\phi\tau - 2b\beta\tau + 2b\gamma^2\phi} \quad (31)$$

An important objective of the supply chain contract is to improve the overall benefit of the supply chain to achieve the effect of centralized control [15]. In order to achieve the maximum profit, contrast $\left(\theta_d^{RSC*}, e_d^{RSC*}, p_d^{RSC*}\right)$ with $(\theta^*, e^*, p^*)$. We find there are no $(w, \phi)$ that can satisfy $\theta_d^{RSC*} = \theta^*$, $e_d^{RSC*} = e^*$ and $p_d^{RSC*} = p^*$ at the same time. $\square$

### 5.2. CS-GS Contract

Since the RSC contract cannot coordinate the supply chain effectively. We propose a new model and try to achieve the supply chain's maximum profit as the centralized model. In the CS-GS model, considering long-term cooperation between the manufacturer and retailer, the retailer shares a part $(1 - l)$ of the R&D costs of the green degree, and the manufacturer shares a part $(f)$ of sales effort costs of advertisement. To encourage the retailer to order more products, the manufacturer wholesale the product at a price $w$. We first determine the contract $(w, l, f)$ that makes sure the decisions made by the manufacturer and the retailer are the same as in the centralized model. The profits of the manufacturer and the retailer are as follows:

$$\pi_m^{CS}(\theta) = (w - c)(z\alpha - \beta p + ke + \gamma\theta) + [(1 - d)p - c][(1 - z)\alpha - \beta(1 - d)p] - \frac{1}{2}l\tau\theta^2 - \frac{1}{2}fbe^2 \quad (32)$$

$$\pi_r^{CS}(e, p) = (p - w)(z\alpha - \beta p + ke + \gamma\theta) - \frac{1}{2}(1 - f)be^2 - \frac{1}{2}(1 - l)\tau\theta^2 \quad (33)$$

**Proposition 5.** *The CS-GS contract can coordinate the supply chain effectively by* $(w, l, f)$ *as follows:*

$$
\begin{cases}
w = \dfrac{\begin{array}{c} \alpha b\gamma^2 + \alpha k^2\tau - \alpha bd\gamma^2 - 4b\beta^2 c\tau - \alpha b\gamma^2 z - \alpha dk^2\tau - \alpha k^2\tau z - 2\alpha b\beta\tau - 2b\beta cd^2\gamma^2 \\ -2\beta cd^2 k^2\tau + 2\alpha b\beta d\tau + 4\alpha b\beta\tau z + 3b\beta cd\gamma^2 + 2b\beta^2 cd\tau + 3\beta cdk^2\tau + \alpha bd\gamma^2 z + \alpha dk^2\tau z \\ +2\alpha b\beta d^2\tau z - 6\alpha b\beta d\tau z \end{array}}{\beta(-2b\beta\tau d^2 + 4b\beta\tau d + b\gamma^2 + \tau k^2 - 4b\beta\tau)} \\[4em]
f = \dfrac{(1-d)\begin{pmatrix} b\beta c\gamma^2 - \alpha k^2\tau - \alpha b\gamma^2 + \beta ck^2\tau + \alpha b\gamma^2 z + \alpha k^2\tau z + 2\alpha b\beta\tau - 4\alpha b\beta\tau z - 2b\beta cd\gamma^2 \\ +2b\beta^2 cd\tau - 2\beta cdk^2\tau + 2\alpha b\beta d\tau z \end{pmatrix}}{b\beta\tau(\alpha - 2\beta c - \alpha d + 3\beta cd + \alpha dz - 2\beta cd^2)} \\[4em]
l = \dfrac{(1-d)\begin{pmatrix} b\beta c\gamma^2 - \alpha k^2\tau - \alpha b\gamma^2 + \beta ck^2\tau + \alpha b\gamma^2 z + \alpha k^2\tau z + 2\alpha b\beta\tau - 4\alpha b\beta\tau z - 2b\beta cd\gamma^2 \\ +2b\beta^2 cd\tau - 2\beta cdk^2\tau + 2\alpha b\beta d\tau z \end{pmatrix}}{b\beta\tau(\alpha - 2\beta c - \alpha d + 3\beta cd + \alpha dz - 2\beta cd^2)}
\end{cases}
\tag{34}
$$

We can find that $l = f$.

**Proof of Proposition 5.** We still formulate the CS-GS model as a Stackelberg game and suppose the retailer is the leader. Firstly, we derive the retailer's reaction function as follows:

$$
\frac{\partial \pi_m^{CS}(\theta)}{\partial \theta} = -\gamma(c - w) - l\tau\theta = 0
$$

$$
\frac{\partial^2 \pi_m^{CS}(\theta)}{\partial \theta^2} = -l\tau < 0
$$

$$
\theta_d^{CS*} = \frac{\gamma(w - c)}{l\tau}
\tag{35}
$$

Substituting Equation (35) into Equation (33), taking the first derivative with respect to $e$ and $p$, we have

$$
\pi_r^{CS}(e, p) = (p - w)\left(z\alpha - \beta p + ke + \frac{\gamma^2(w - c)}{l\tau}\right) - \frac{1}{2}(1 - f)be^2 - \frac{1}{2}(1 - l)\tau\left(\frac{\gamma(w - c)}{l\tau}\right)^2
\tag{36}
$$

Taking the first derivative of Equation (36) with respect to $e$ and $p$, we obtain:

$$
\frac{\partial \pi_r^{CS}(e, p)}{\partial e} = k(p - w) - b(1 - f)e
$$

$$
\frac{\partial \pi_r^{CS}(e, p)}{\partial p} = ek - \beta p + \alpha z - \beta(p - w) - \frac{\gamma^2(c - w)}{l\tau}
$$

Then the Hessian matrix is

$$
H_2 = \begin{bmatrix} \dfrac{\partial^2 \pi_r^{rd}(e, p)}{\partial e^2} & \dfrac{\partial^2 \pi_r^{rd}(e, p)}{\partial e\partial p} \\[1.5em] \dfrac{\partial^2 \pi_r^{rd}(e, p)}{\partial p\partial e} & \dfrac{\partial^2 \pi_r^{rd}(e, p)}{\partial p^2} \end{bmatrix} = \begin{bmatrix} -b(1 - f) & k \\ k & -2\beta \end{bmatrix}
$$

It is obvious that when $2b\beta(1 - f) - k^2 > 0$, the Hessian matrix is a negatively defined matrix, which implies that $\pi_r^{CS}(e, p)$ is jointly concave in $(e_d^{CS*}, p_d^{CS*})$, and the maximum $(e_d^{CS*}, p_d^{CS*})$ is uniquely solved by

$$
k(p - w) - b(1 - f)e = 0
$$

$$
ek - \beta p + \alpha z - \beta(p - w) - \frac{\gamma^2(c - w)}{l\tau} = 0
$$

The manufacturer's optimal green degree of the product, the retailer's optimal price and optimal sales effort, denoted as $(\theta_d^{CS*}, e_d^{CS*}, p_d^{CS*})$, are

$$
\theta_d^{CS*} = \frac{\gamma(w - c)}{l\tau}
\tag{37}
$$

$$
e_d^{CS*} = \frac{k(c\gamma^2 - \gamma^2 w + \beta l\tau w - \alpha l\tau z)}{l\tau(k^2 - 2b\beta + 2b\beta f)}
\tag{38}
$$

$$p_d^{CS*} = \frac{bc\gamma^2 - b\gamma^2 w - bcf\gamma^2 + bf\gamma^2 w + k^2 l\tau w - b\beta l\tau w - \alpha bl\tau z + b\beta fl\tau w + \alpha bfl\tau z}{l\tau(k^2 - 2b\beta + 2b\beta f)} \tag{39}$$

In the CS-GS model, the supply chain should be coordinated when $\theta_d^{CS*} = \theta^*$, $e_d^{CS*} = e^*$ and $p_d^{CS*} = p^*$. We find that when $(w, l, f)$ satisfy Equation (34), the optimal decisions are the same as that under the centralized model. It can prove that the contract we proposed can achieve the maximum profit for the entire supply chain and we obtain:

$$\pi_m^{CS*} = \pi_m^{CS}(\theta_d^{CS*}) = \max\pi_m^{CS}(\theta)$$
$$\pi_r^{CS*} = \pi_m^{CS}(e_d^{CS*}, p_d^{CS*}) = \max\pi_m^{CS}(e, p)$$
$$\pi_m^{CS*} + \pi_r^{CS*} = \pi_c^*$$

□

It is proved that CS-GS contracts can effectively coordinate the supply chain. The next question is how to encourage the manufacturer and the retailer to adopt the contract. The members of the supply chain are usually independent. It is obvious that the members will participate in the coordination mechanism only if all the members can obtain extra profits from the cooperation. To solve this question, we use Nash bargaining analysis, which provides $T$ (a negotiated value that is transferred from the manufacturer to the retailer) leading to a win-win situation. Here, the cooperation parameter $\xi_i$ represents members' bargaining power or its importance in cooperation.

$$\pi_{mT}^{CS} = \pi_m^{CS*} - T \tag{40}$$

$$\pi_{rT}^{CS} = \pi_r^{CS*} + T \tag{41}$$

When $\pi_m^{CS*} - \pi_m^{rd*} > 0$ and $\pi_r^{CS*} - \pi_r^{rd*} < 0$, $T$ is positive. When $\pi_m^{CS*} - \pi_m^{rd*} < 0$ and $\pi_r^{CS*} - \pi_r^{rd*} > 0$, $T$ is negative. The value of $T$ depends on the negotiation power between the manufacturer and the retailer, and it satisfies $\pi_r^{rd*} - \pi_r^{CS*} < T < \pi_m^{CS*} - \pi_m^{rd*}$, which makes sure that the manufacturer and the retailer can obtain more profit from their cooperation.

In the RD model, for example, the extra profits of the manufacturer and the retailer compared to the decentralized model are:

$$\Delta\prod_m = \pi_{mT}^{CS} - \pi_m^{rd*}$$
$$\Delta\prod_r = \pi_{rT}^{CS} - \pi_r^{rd*}$$

We use the Nash bargaining model to identify the optimal profit allocation solution. We assume that the utility functions of the manufacturer and the retailer are $u_m(T) = \Delta\prod_m{}^{\xi_1}$, $u_r(T) = \Delta\prod_r{}^{\xi_2}$ and $\sum_{i=1}^2, \xi_i = 1(i = 1, 2)$. Using profits of each supply member under the decentralized model as the status quo, the problem is then to find $T^*$ that maximizes

$$N(T) = [u_m(T)]^{\xi_1}[u_r(T)]^{\xi_2} \tag{42}$$

Let the first-order conditions and second-order conditions of $N(T)$ with respect to $T$,

$$\frac{dN(T)}{dT} = \xi_2\left(\pi_m^{CS*} - \pi_m^{rd*} - T\right)^{\xi_1}\left(T - \pi_r^{rd*} + \pi_r^{CS*}\right)^{\xi_2-1} - \xi_1\left(\pi_m^{CS*} - \pi_m^{rd*} - T\right)^{\xi_1-1}\left(T - \pi_r^{rd*} + \pi_r^{CS*}\right)^{\xi_2}$$

$$= \frac{(1-\xi_1)\left(\pi_m^{CS*} - \pi_m^{rd*} - T\right)^{\xi_1}}{\left(T - \pi_r^{rd*} + \pi_r^{CS*}\right)^{\xi_1}} - \frac{\xi_1\left(T - \pi_r^{rd*} + \pi_r^{CS*}\right)^{1-\xi_1}}{\left(\pi_m^{CS*} - \pi_m^{rd*} - T\right)^{1-\xi_1}}$$

$$\frac{dN^2(T)}{dT^2} = -\frac{\xi_1(1-\xi_1)\left(\pi_m^{CS*} - \pi_m^{rd*} - T\right)^{\xi_1}}{\left(T - \pi_r^{rd*} + \pi_r^{CS*}\right)^{\xi_1+1}} - \frac{2\xi_1(1-\xi_1)}{\left(T - \pi_r^{rd*} + \pi_r^{CS*}\right)^{\xi_1}\left(\pi_m^{CS*} - \pi_m^{rd*} - T\right)^{1-\xi_1}}$$
$$- \frac{\xi_1(1-\xi_1)\left(T - \pi_r^{rd*} + \pi_r^{CS*}\right)^{1-\xi_1}}{\left(\pi_m^{CS*} - \pi_m^{rd*} - T\right)^{2-\xi_1}}$$

It is obvious that $\frac{dN^2(T)}{dT^2} < 0$, so there is a unique $T$ that maximizes $N(T)$, when $T = (1-\xi_1)(\pi_m^{CS*} - \pi_m^{rd*}) + \xi_1(\pi_r^{rd*} - \pi_r^{CS*})$, $\frac{dN(T)}{dT} = 0$, and the Nash bargaining model achieves optimally.

**Proposition 6.** *In the Nash bargaining model, when $T = (1 - \xi_1)(\pi_m^{CS*} - \pi_m^{rd*}) + \xi_1(\pi_r^{rd*} - \pi_r^{CS*})$, $\xi_1$ represents the manufacturer's bargaining power, the members' utility can achieve optimally, and they can both achieve more profits from the cooperation.*

## 6. Analysis

In this section, numerical analysis is given to illustrate the effectiveness of the proposed models; moreover, the influence of parameters on optimal decisions and profits is discussed. For ease of analysis, we summarize the decisions made by members of the supply chain under different models in Table 1.

### 6.1. The Impact of Coordination Contract on the Supply Chain's Performance

For numerical study, we assume $\alpha = 400$, $\beta = 0.6$, $k = 0.8$, $\gamma = 0.5$, $z = 0.3$, $d = 0.2$, $\tau = 4$, $b = 5$, $w = 40$, $c = 30$ and $\xi_1 \in (0, 1)$. The contract parameters, optimal decisions and profits are calculated in Table 2. As we can see, compared to the decisions made in the centralized model, retail price, green degree and sales effort decisions are all lower in the two decentralized models, and thus the profit of the whole supply chain is lower in the two decentralized models too. We find that the profits of the whole supply chain cannot achieve optimally despite whether the manufacturer or the retailer is dominant in the supply chain. According to the calculation results, the profits of the manufacturer and the retailer are both increased by RSC, but the whole profits of the supply chain are still less than that in the centralized model. In the CS-GS coordination model, the manufacturer sells the product to the retailer at 160.4, the retailer shares 27% of the R&D costs for the green degree, and the manufacturer shares 73% of sales effort costs for the advertisements, resulting in the optimal decisions and profits being the same as those in the centralized model. When $\xi_1 = 0.6$, the profits of the manufacturer and the retailer are both more than those in the decentralized model without coordination, which motivates the manufacturer and the retailer to participate in this coordination contract mechanism.

As Figure 1 shows, the profit-transferring from the manufacturer to the retailer decreases from 8935.05 to 4354.58 as the manufacturer's bargaining power increases from 0.1 to 0.9, and $T$ satisfies $3782.1 < T < 9507.6$ ($\pi_r^{rd*} - \pi_r^{CS*} < T < \pi_m^{CS*} - \pi_m^{rd*}$) all the time. For convenience to study, the maximum profits of the manufacturer and the retailer without coordination are shown together in Figure 1. It reveals that the optimal profit of the manufacturer increased and that the optimal profit of the retailer decreased with the manufacturer's bargaining power increase, and they are both more than that in the decentralized model without a coordination mechanism.

**Table 1.** Summary of the optimal decisions.

| Optimal Decisions | Green Degree ($\theta$) | Sales Effort ($e$) | Retail Price ($p$) |
|---|---|---|---|
| Centralized model | $\dfrac{k\tau\left(2\beta c-\alpha+\alpha d-3\beta cd-\alpha dz+2\beta cd^2\right)}{-2b\tau\beta d^2+4b\tau\beta d+\tau k^2+b\gamma^2-4b\tau\beta}$ | $\dfrac{b\gamma\left(2\beta c-\alpha+\alpha d-3\beta cd-\alpha dz+2\beta cd^2\right)}{-2b\tau\beta d^2+4b\tau\beta d+\tau k^2+b\gamma^2-4b\tau\beta}$ | $\dfrac{c\tau k^2+bc\gamma^2-\alpha b\tau-2bc\tau\beta+\alpha bd\tau-\alpha bd\tau z+bcd\tau\beta}{-2b\tau\beta d^2+4b\tau\beta d+\tau k^2+b\gamma^2-4b\tau\beta}$ |
| Decentralized model (MD) | $\dfrac{b\beta\gamma(c-w)}{\sigma_1}-\dfrac{b\gamma(d-1)\left(\alpha(z-1)-\frac{\beta(d-1)\sigma_2}{\sigma_1}\right)}{\sigma_1}$ $-\dfrac{+\dfrac{b\beta\gamma\left(c+\frac{(d-1)\sigma_2}{\sigma_1}\right)(d-1)}{\sigma_1}}{\tau+\frac{2b^2\beta\gamma^2(d-1)^2}{\sigma_1^2}}$ $\sigma_1=2b\beta-k^2$ $\sigma_2=-wk^2+b\beta w+\alpha bz$ | $\dfrac{k\left(\alpha z-\beta w+\theta_d^{md*}\gamma\right)}{2\beta b-k^2}$ | $\dfrac{zb\alpha+b\theta_d^{md*}\gamma+b\beta w-k^2 w}{2\beta b-k^2}$ |
| Decentralized model (RD) | $\dfrac{\gamma(w-c)}{\tau}$ | $\dfrac{k\left(-c\gamma^2+\gamma^2 w-\beta\tau w+\alpha\tau z\right)}{\tau(2b\beta-k^2)}$ | $\dfrac{b\gamma^2 w-bc\gamma^2-k^2\tau w+b\beta\tau w+\alpha b\tau z}{\tau(2b\beta-k^2)}$ |
| Coordination model (RSC) | $-\gamma\dfrac{\begin{pmatrix}ck^2\tau-k^2\tau w+2b\beta\tau w+bc\gamma^2\phi-b\gamma^2\phi w\\-ck^2\phi\tau+k^2\phi\tau w-2b\beta c\tau-b\beta\phi\tau w\\+\alpha b\phi\tau z\end{pmatrix}}{\tau(k^2\tau-k^2\phi\tau-2b\beta\tau+2b\gamma^2\phi)}$ | $\dfrac{k(1-\phi)\left(c\gamma^2-\gamma^2 w+\beta\tau w-\alpha\tau z\right)}{k^2\tau-k^2\phi\tau-2b\beta\tau+2b\gamma^2\phi}$ | $\dfrac{\begin{array}{c}-b\gamma^2 w+bc\gamma^2+k^2\tau w-b\beta\tau w-\alpha b\tau z\\+2b\gamma^2\phi w-k^2\phi\tau w\end{array}}{k^2\tau-k^2\phi\tau-2b\beta\tau+2b\gamma^2\phi}$ |
| Coordination model (CS-GS) | $\dfrac{\gamma(w-c)}{l\tau}$ | $\dfrac{k\left(c\gamma^2-\gamma^2 w+\beta l\tau w-\alpha l\tau z\right)}{l\tau(k^2-2b\beta+2b\beta f)}$ | $\dfrac{\begin{array}{c}bc\gamma^2-b\gamma^2 w-bcf\gamma^2+bf\gamma^2 w+k^2 l\tau w\\-b\beta l\tau w-\alpha bl\tau z+b\beta fl\tau w+\alpha bfl\tau z\end{array}}{l\tau(k^2-2b\beta+2b\beta f)}$ |

<p style="text-align:center">Table 2. The optimal decisions and profits under different conditions.</p>

| Model | Coordination Parameters | $\theta$ | $e$ | $p$ | $\pi_m$ | $\pi_r$ | $\pi$ |
|---|---|---|---|---|---|---|---|
| Centralized model | - | 22.3 | 28.6 | 208.5 | - | - | 26,737.6 |
| Decentralized model (MD) | - | 16.2 | 15.5 | 137.1 | 17,125.7 | 5055.4 | 22,181.1 |
| Decentralized model (RD) | - | 1.25 | 14.4 | 130.1 | 16,658.8 | 4354.7 | 21,013.5 |
| Coordination model (RSC) | $(w, \phi)$ (197.3, 0.6) When $\xi_1 = 0.6$ | 0.72 | 21.8 | 208.5 | 18,929.8 | 5868.5 | 24,798.3 |
| Coordination model (CS-GS) | $(w, l, f)$ (160.4, 0.73, 0.73) When $\xi_1 = 0.6$ | 22.3 | 28.6 | 208.5 | 20,093.4 | 6644.2 | 26,737.6 |

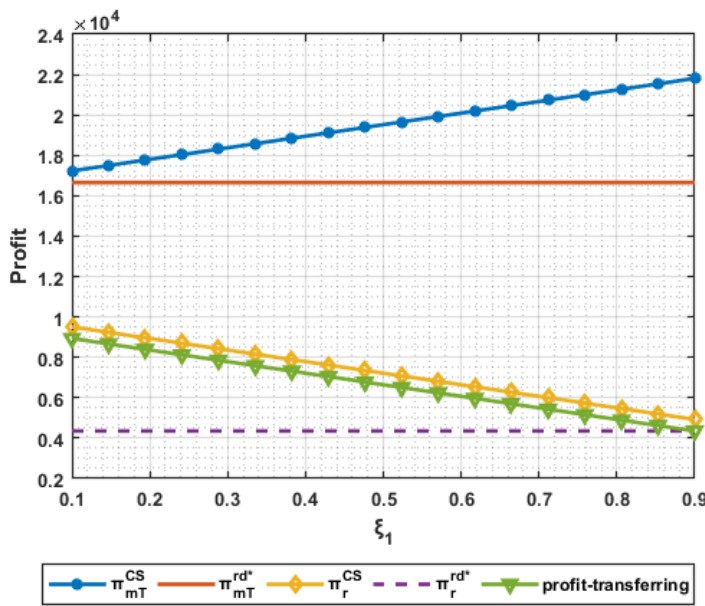

**Figure 1.** The impact of $\xi_1$ on the profit of supply chain members.

### 6.2. The Impacts of $\tau$ and $b$ on the Profits of Supply Chain and Members

In this section, we assume $\alpha = 400$, $\beta = 0.6$, $k = 0.8$, $\gamma = 0.5$, $z = 0.3$, $d = 0.2$, $c = 30$, $\tau$ and $b$ changes from 0.75 to 5, which all meet the assumptions mentioned before. Figure 2 illustrates that the supply chain's profit under the CS-GS model is always higher than that in the RSC model with the changes of $\tau$ and $b$. When $\tau$ and $b$ are both bigger than 2, the supply chain's profit becomes stable, and the profit gap between the two models decreases with the increase of $\tau$ and $b$. Figure 3 shows that the optimal profits of the supply chain and members all decrease with $\tau$ increases, especially when the manufacturer is dominant in the supply chain. As we can see from Figure 4, when $b$ increases, the optimal profits of the supply chain and members all decrease. Obviously, the manufacturer will reduce the investment in the green degree when $\tau$ is increased, and the retailer will reduce the investment in the sales effort when $b$ increases.

Figures 3 and 4 show that the supply chain's optimal profit under the CS-GS model can always reach the integrated benchmark when $\tau$ and $b$ change from 0.75 to 5; however, both the profits of the supply chain in the MD model or in the RD model cannot achieve maximum. In addition, the total profits of the supply chain in the MD model are always higher than that in the RD model, and the manufacturer's profit is higher than the retailer's. It is verified that there is always an imbalance in power in the supply chain; therefore, the members' relationship cannot remain stable. Without an appropriate coordination mechanism, the supply chain members make decisions from their own

standpoints without considering the whole interest, which causes double marginalization, and the CS-GS contract we proposed can help them obtain the maximum profits.

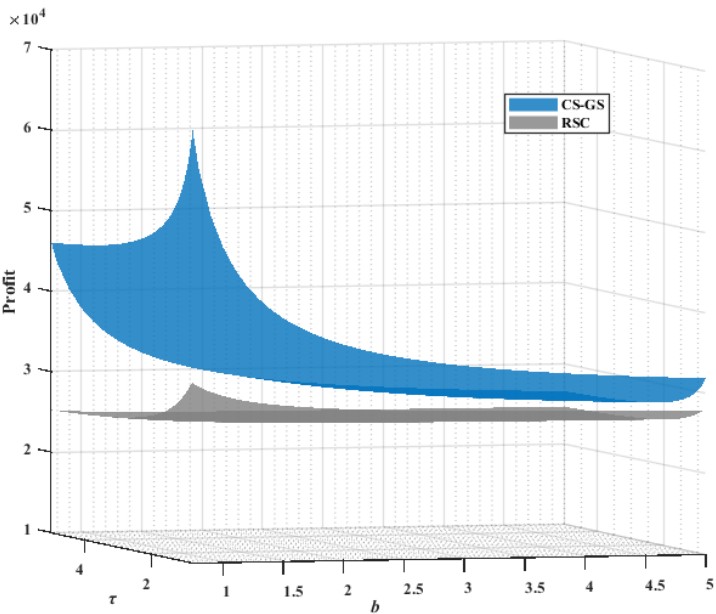

**Figure 2.** The joint impacts of $\tau$ and $b$ on the supply chain's profits.

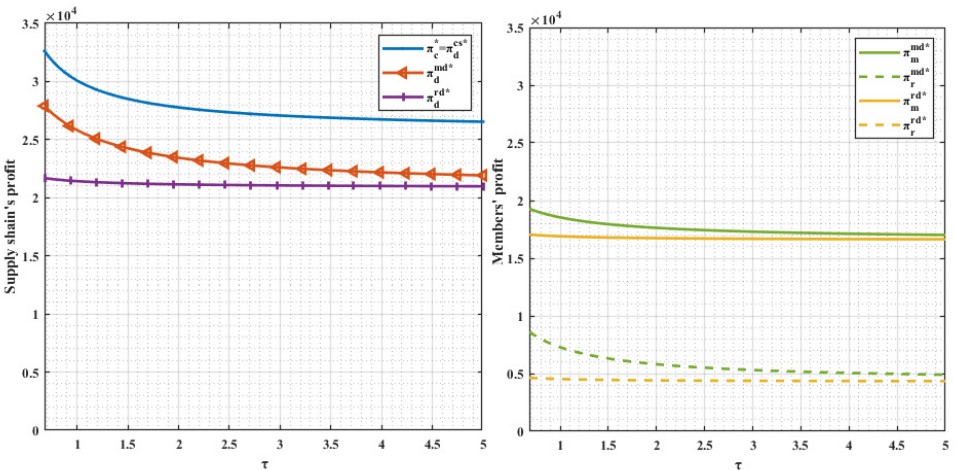

**Figure 3.** The impact of $\tau$ on the profits of supply chain and members.

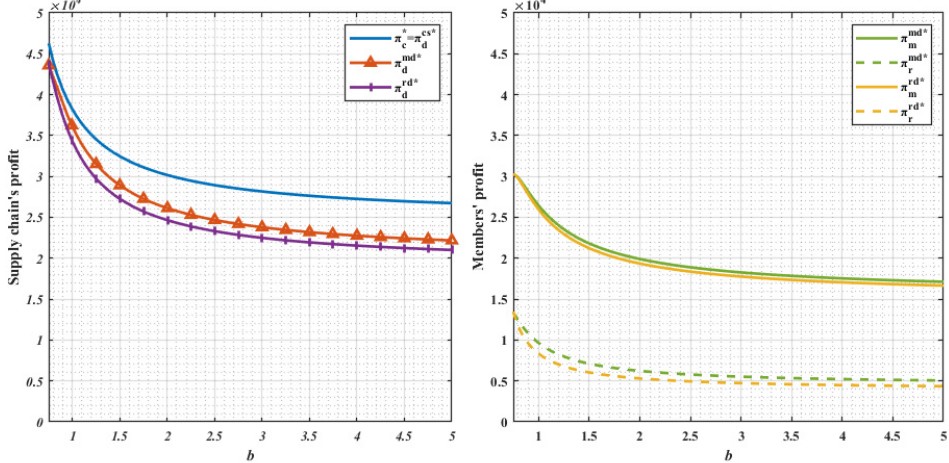

**Figure 4.** The impact of $b$ on the profits of supply chain and members.

*6.3. The Impacts of Relevant Parameters on the Decisions*

6.3.1. The Impacts of $\tau$ and $\gamma$ on the Green Degree of the Product

We employ the same parameters as those in Section 6.1: $\alpha = 400$, $\beta = 0.6$, $k = 0.8$, $b = 5$, $z = 0.3$, $d = 0.2$, $w = 40$, and $c = 30$. Figure 5 illustrates the impact of $\tau$ on the green degree of the product in four conditions. It is shown that the green degree of the product is decreased with $\tau$ increases in these four models, and the difference in green degree in the four models decreases with $\tau$ increases. We can also see that the optimal green degree of the product first decreases fast and then decreases slowly in the centralized and CS-GS models. In the decentralized model, the retailer will not pay the R&D costs of a green degree, the green degree of the product changes a little with $\tau$ increases when the retailer is dominant. The green degree of the product in the CS-GS coordination model is always more than that in the decentralized model, so the CS-GS coordination contract can promote the green degree of the product effectively.

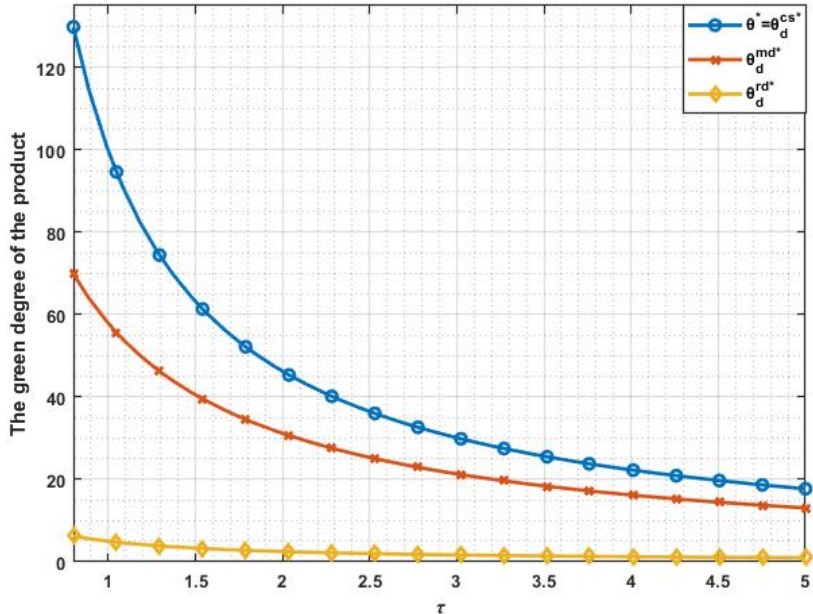

**Figure 5.** The impact of $\tau$ on the green degree of the product.

As Figure 6 shows, in the centralized model, the green degree of the product is the highest. In the decentralized model without coordination, the green degree of the product in the MD model is more than that in the RD model. Through the CS-GS coordination contract, the green degree of the product can be improved significantly. With $\gamma$ increases, the green degree of the product is increased in four conditions, especially in the centralized model and coordination model. It is because the increase of $\gamma\theta$ can improve consumer demand (with the consumer demand increasing, the profit of the supply chain will increase too). Therefore, when the green degree of the product gains more consumers' preference, the CS-GS coordination contract can help consumers to react faster to this compared to other decentralized models without coordination.

6.3.2. The Impacts of $b$ and $k$ on the Sales Effort of the Advertisement

Assume $\alpha = 400$, $\beta = 0.6$, $\gamma = 0.5$, $\tau = 4$, $z = 0.3$, $d = 0.2$, $w = 40$, $c = 30$. In this section, we analyze the impacts of $b$ and $k$ on the sales effort of the advertisement in four models. As shown in Figure 7, the sales effort of the advertisement decreases when $b$ increases in four conditions because the increase of $b$ leads to high cost, and then the demand and profits will be reduced as shown in Figure 4 before. Similar to Figure 6, we can also find that when $k$ increases, the sales effort of the advertisement increases, leading to higher demand and profit, which is in line with common sense. From Figures 7 and 8, we can draw some conclusions. When $b$ increases from 1 to 5 and $k$ increases from 0 to 1, the sales effort of the advertisement in the centralized and CS-GS model is always the highest, and in the MD model, there is little more than that which is in the RD model.

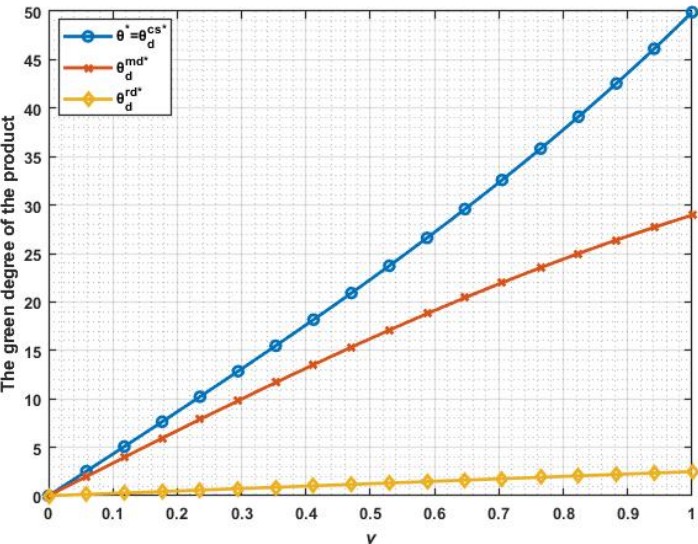

**Figure 6.** The impact of $\gamma$ on the green degree of the product.

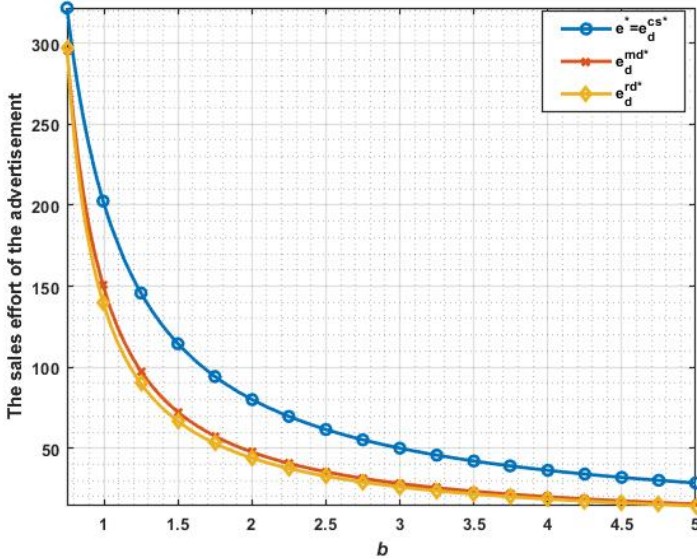

**Figure 7.** The impact of $b$ on the sales effort of the advertisement.

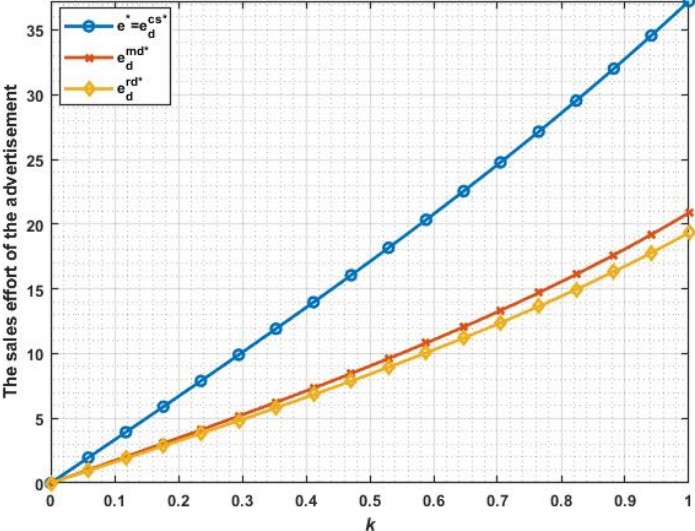

**Figure 8.** The impact of $k$ on the sales effort of the advertisement.

### 6.3.3. The Impacts of $d$ and $\beta$ on the Price

Assume $\alpha = 400$, $k = 0.8$, $\gamma = 0.5$, $z = 0.3$, $\tau = 4$, $b = 5$, $w = 40$, $c = 30$, $d$ changes from 0.1 to 0.9, $\beta$ changes from 0.5 to 1. Figure 9 shows that, in the centralized and CS-GS model, the retail price increases when the discount of the retail price in the live streaming rooms increases from 0.1 to 0.427, and the retail price will be decreased when the discount increases from 0.427 to 0.9. When the discount of the retail price in the live streaming rooms equals 0.427, the retail price achieves its maximum. We can also find that the retail price in the decentralized models is lower than that in the centralized and CS-GS models. Moreover, when the discount of the retail price in the live streaming rooms increases, the retail price in the MD model decreases gradually, but it is a fixed value in the RD model. It reveals that the price decision made by the retailer in the RD model will not be influenced by the discount for the manufacturer's live streaming rooms.

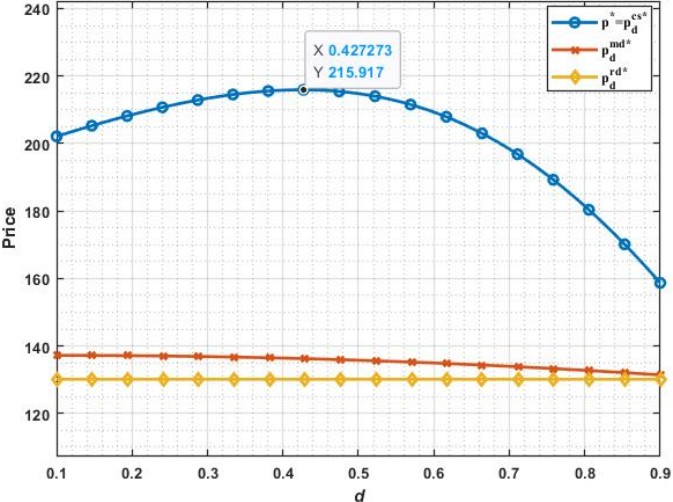

**Figure 9.** The impact of $d$ on price.

Figure 10 shows the impact of $\beta$ on the retail price in four models. The retail price in the centralized and CS-GS models is the highest, and it is decreased with $\beta$ increases. It is congenial with reason and common sense. We can also find that when $\beta$ increases from 0.5 to 1, the retail price under the two decentralized models is lower than that in the centralized model. It is indicated that the manufacturer and the retailer both tend to stimulate demand by driving a lower price. Therefore, the profit of the supply chain under the decentralized model without coordination contracts cannot reach the maximum value. However, the CS-GS coordination contract can help to improve the retail price, leading to higher profit.

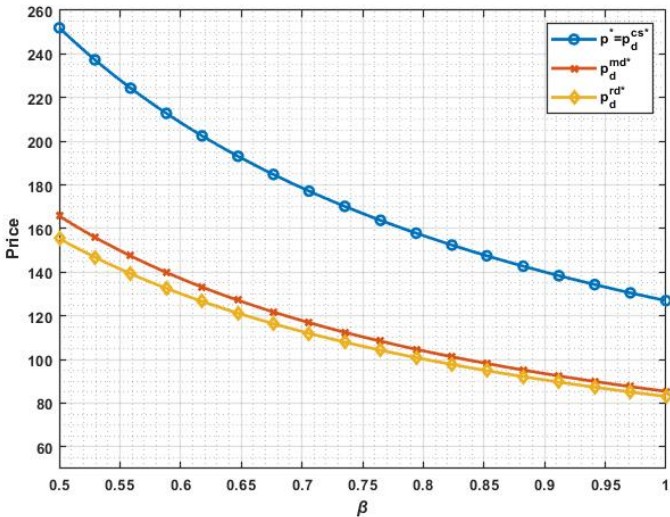

**Figure 10.** The impact of $\beta$ on price.

## 7. Conclusions

With rising environmental awareness among consumers, the demand for green products has increased and green product sales via live streaming rooms are becoming more and more popular, especially in the post-COVID-19 business environment. This paper investigates a two-echelon green supply chain, including a manufacturer and a retailer, and analyzes the price, sales effort and green degree decisions made by the retailer and manufacturer. Different from the conventional studies, this paper considers a new type of selling channel called live streaming rooms besides the traditional channel. The demand for live-streaming rooms is dependent on retail prices and discounts due to the characteristic of target consumers. Referring to previous research, the demand in the traditional channel is a linear function of the retail price, green degree and sales effort.

Several decision models are developed to determine the optimal price, sales effort of the advertisement and green degree of the product: (1) centralized model, (2) manufacturer dominant decentralized model, (3) retailer dominant decentralized model, (4) RSC coordination model and (5) CS-GS coordination model. By comparison with optimal decisions and profits in these models. It is found that the commonly used RSC contract cannot coordinate the supply chain completely, but the CS-GS coordination contract can effectively motivate the supply chain members to make the same optimal decisions as that in the centralized model, so as to achieve the maximum profits of the entire system. In addition, each member in the supply chain can obtain more profits by extra profits allocation through bargaining problems, which verifies the feasibility of the CS-GS coordination contract achieving a win-win situation. All models are evaluated by numerical analysis, and this reveals that the CS-GS coordination contract can help to improve the green degree of the products, which is beneficial to our consumers and the environment. Moreover, when the supply chain members cooperate by CS-GS coordination contract, the discount in the live streaming rooms is not always beneficial to the whole supply chain's interests, which means the manufacturer should make an appropriate discount for the live streaming anchors.

For further research, we propose that the supply chain with multiple manufacturers and multiple retailers can be studied, and the model over multiple periods can also be considered in the future. Moreover, the third party can join the supply chain, such as the government paying for some green investment in the form of subsidies, the logistics enterprise responsible for recycling, and so on. Dynamic and stochastic factors can also be used in the demand function to analyze a more complex market environment. Some other coordination mechanisms can be proposed in further study. Even if the optimal coordination state cannot be achieved, it can also achieve the Pareto optimum.

**Author Contributions:** Methodology, T.C.; Formal analysis, T.C.; Funding acquisition, T.C.; Writing—review and editing, T.C.; Methodology, R.Z.; Formal analysis, R.Z.; Investigation, C.L.; Formal analysis, C.L.; Funding acquisition, C.L.; Methodology, X.X.; Writing—review and editing, X.X. All authors have read and agreed to the published version of the manuscript.

**Funding:** This research was funded by Yancheng Industry Education Integration Development Research Center, (2022cjrh010); School-level research projects of Yancheng Institute of Technology (xjr2020026); Senior Talent Research Project of Yancheng Polytechnic College, (6070017/007).

**Institutional Review Board Statement:** Not appliable.

**Informed Consent Statement:** Not appliable.

**Data Availability Statement:** Not appliable.

**Conflicts of Interest:** The authors declare no conflict of interest.

### Nomenclature

Decision variables:
$p$    the unit product's retail price;
$\theta$    the green degree of the product;
$e$    the sales effort of the advertisement;
$D$    consumer's demand, which is a function of $p$, $\theta$, $e$, $d$;

Parameters:

$c$    the unit manufacturing cost;
$\tau$    the coefficient of the green investment;
$b$    the scale parameter of the advertisement;
$w$    the unit wholesale price;
$\alpha$    the market base of this product;
$\beta$    the responsiveness of the consumer demand to retail price;
$k$    the responsiveness of the consumer demand to the sales effort;
$r$    the responsiveness of the consumer demand to the green degree;
$\widetilde{p}$    the selling price in live streaming rooms
$d$    the discount of the retail price in live streaming rooms

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
