# Peer review of "Research on Coordination in a Dual-Channel Green Supply Chain under Live Streaming Mode"

_sustainability, doi:10.3390/su15010878_

Round 1

Reviewer 1 Report

Thank you very much for allowing me to review your work. This work shows that there is effort and perseverance in its preparation. After a thorough review, some errors or mistakes have been detected and should be revised. If you had used the review format proposed by the journal, the review would have been easier. I will comment on the different aspects to be taken into account:

The decisions taken at the time of the modernisation being analysed should be well justified and not respond to subjective assumptions of the researchers.

Figures 3, 4 and 5 provide a lot of information that is not easy for the reader to understand. In particular, figure 3, being a 3D graph, there is information that is lost and overlaps with some data and variables. On the other hand, figure 1 provides very little information to the reader, it would be convenient to include both in this figure and in figure 2 the values of the variables in the graph.

After analysing the results obtained, there is a lack of a discussion that relates the results to the theoretical framework described.

Furthermore, the conclusions drawn are superficial, and could go into more depth on the basis of the different variables analysed.

I reiterate my thanks and look forward to reading your reviews in the near future.

Best regards,

Author Response

Thank you very much for allowing me to review your work. This work shows that there is effort and perseverance in its preparation. After a thorough review, some errors or mistakes have been detected and should be revised. If you had used the review format proposed by the journal, the review would have been easier. I will comment on the different aspects to be taken into account:

  1. The decisions taken at the time of the modernisation being analysed should be well justified and not respond to subjective assumptions of the researchers. Figures 3, 4 and 5 provide a lot of information that is not easy for the reader to understand. In particular, figure 3, being a 3D graph, there is information that is lost and overlaps with some data and variables. On the other hand, figure 1 provides very little information to the reader, it would be convenient to include both in this figure and in figure 2 the values of the variables in the graph. After analysing the results obtained, there is a lack of a discussion that relates the results to the theoretical framework described.

Response: Thanks for the reviewer’s comments. It is revised in the manuscript.

Figure 1 and figure 2 are combined as Figure 1 shown in the revised manuscript, and it reveals that the optimal profit of the manufacturer is increased and the optimal profit of the retailer is decreased with the manufacturer’s bargaining power increases, and they are both more than that in the decentralized model without coordination mechanism.

Figure 3 is adjusted clearly as Figure 2 shown in the revised manuscript.

Figures 4, 5 are adjusted as Figure 3, 4 shown in the revised manuscript, in which we discuss the result in detail and the revised parts are displayed in the revised manuscript.

  1. Furthermore, the conclusions drawn are superficial, and could go into more depth on the basis of the different variables analysed.

Response: The analysis of different variables in the conclusions is revised as ‘By comparison with optimal decisions and profits in these models. It is found that the commonly used RSC contract cannot coordinate the supply chain completely, but CS-GS coordination contract can effectively motivate the supply chain members to make the same optimal decisions as that in the centralized model, so as to achieve the maximum profits of the entire system. In addition, each member in the supply chain can obtain more profits by extra profits allocation though bargaining problem, which verifies the feasibility of CS-GS coordination contract achieving a win-win situation. All models are evaluated by numerical analysis, it reveals that CS-GS coordination contract can help to improve green degree of the products, which is benefit to our consumers and the environment. Moreover, when the supply chain members cooperate by CS-GS coordination contract, the discount in the live streaming rooms isn’t always beneficial to the whole supply chain’s interests, which means the manufacturer should make an appropriate discount for the live streaming anchors.’

Reviewer 2 Report

Thank you for the opportunity to cooperate with Sustainability (MDPI), collaborating to improve the manuscript entitled "Research on coordination in a dual-channel green supply chain under live streaming mode", which aims to study the coordination issue in a dual-channel green supply chain with one manufacturer and one retailer.

Congratulations to the authors for their hard work.

a) Name o the authors: ..., Ronghu zhou1, ... - Please write your family name starting with a capital letter.

b) Abstract -Please state the meaning the first time an abbreviation is presented, such as ... "CS-GS" in the Abstract. Please inform the methodology used.

1) Introduction: The context was described, presenting previous and recent theoretical background on the topic.

2) Related literature: Well conducted.

Please avoid long paragraphs.

Please, why the sentence that follows was written in a reduced font size?

...analyzed a two-stage Stackelberg game model, in which both the demand and yield are uncertain. They studied and compared the production and ordering decisions under two different models, and proposed a subsidy mechanism to achieve Pareto-improvement.

3. Model descriptions, notations and the integrated benchmark - It was substantively explained.

4. The decentralized model - The formulas were developed.

5. Supply chain coordination - The proof of the propositions were demonstrated.

6. Analysis -  Numerical analysis was given to present the effectiveness of proposed models and the influence of parameters on the optimal decision and profit of dual-channel supply chain members was discussed. Well structured.

7. Conclusions - Please write "in" in lower letter, in the sentence: ...more and more popular especially In the Post-COVID-19...

1) Please correct thess phrases:

a) For further research, this paper studies a two-echelon supply chain involving a manufacturer and a retailer, the supply chain with multiple..

or propose studies?

b) We can also further study some

or propose further...

8. References

Please the name of the journals must be written in an abbreviated form.

The year must be inserted after the name of the journal.

Please see the journal's template.

Author Response

Thank you for the opportunity to cooperate with Sustainability (MDPI), collaborating to improve the manuscript entitled "Research on coordination in a dual-channel green supply chain under live streaming mode", which aims to study the coordination issue in a dual-channel green supply chain with one manufacturer and one retailer.

Congratulations to the authors for their hard work.

  1. Name o the authors: ..., Ronghu zhou1, ... - Please write your family name starting with a capital letter.

Response: Thanks for the reviewer’s comments. It is revised in the manuscript.

  1. Abstract -Please state the meaning the first time an abbreviation is presented, such as ... "CS-GS" in the Abstract. Please inform the methodology used.

Response: It is revised as ‘sharing R&D costs of green degree and sales effort costs of advertisement (CS-GS)’ in the abstract.

  1. Introduction: The context was described, presenting previous and recent theoretical background on the topic.

Related literature: Well conducted.

Please avoid long paragraphs.

Response: The parts of long paragraphs are revised in the related literature.

  1. Please, why the sentence that follows was written in a reduced font size?

...analyzed a two-stage Stackelberg game model, in which both the demand and yield are uncertain. They studied and compared the production and ordering decisions under two different models, and proposed a subsidy mechanism to achieve Pareto-improvement.

Response: There is an error in font size, and it is revised.

  1. Model descriptions, notations and the integrated benchmark - It was substantively explained. The decentralized model - The formulas were developed. Supply chain coordination - The proof of the propositions were demonstrated. Analysis - Numerical analysis was given to present the effectiveness of proposed models and the influence of parameters on the optimal decision and profit of dual-channel supply chain members was discussed. Well structured.

Response: Thanks for the reviewer’s comments.

  1. Conclusions - Please write "in" in lower letter, in the sentence: ...more and more popular especially In the Post-COVID-19...

Response: Done as suggestion.

  1. Please correct thess phrases:
  2. a) For further research, this paper studies a two-echelon supply chain involving a manufacturer and a retailer, the supply chain with multiple.

or propose studies?

  1. b) We can also further study some

or propose further...

Response: It is revised as ‘For further research, we propose that the supply chain with multiple manufacturers and multiple retailers can be studied, and the model over multiple periods can also be considered in the future. Moreover, the third party can join the supply chain, such as the government paying for some green investment in the form of subsidies, the logistics enterprise responsible for recycling, and so on. Dynamic and stochastic factors can also be used in the demand function to analyze more complex market environment. Some other coordination mechanisms can be proposed in further study. Even if the optimal coordination state cannot be achieved, it can also achieve the Pareto optimum.’

  1. References

Please the name of the journals must be written in an abbreviated form. The year must be inserted after the name of the journal. Please see the journal's template.

Response: It is revised in the manuscript.

Round 2

Reviewer 1 Report

After the different contributions given in the first revision, it is appreciated that most of the suggestions made have been taken into account. The increase in the quality of the manuscript is noticeable.